# The Soundscape Hackathon as a Methodology to Accelerate Co-Creation of the Urban Public Space

**Jorg De Winne** [1,2,*], **Karlo Filipan** [3], **Bart Moens** [2], **Paul Devos** [1], **Marc Leman** [2],
**Dick Botteldooren** [1] and **Bert De Coensel** [1,3]

1    WAVES Research Group, Department of Information Technology, Ghent University, 9000 Ghent, Belgium;
     p.devos@ugent.be (P.D.); Dick.Botteldooren@UGent.be (D.B.); bert.decoensel@ugent.be (B.D.C.)
2    IPEM, Department of Art, Music and Theater Studies, Ghent University, 9000 Ghent, Belgium;
     bart.moens@ugent.be (B.M.); marc.leman@ugent.be (M.L.)
3    ASAsense cvba, 8000 Bruges, Belgium; karlo.filipan@asasense.com
*    Correspondence: jorg.dewinne@ugent.be

**Abstract:** The design of urban public spaces is typically performed by architects and urban planners, which often only focus on the visual aesthetics of the urban space. Yet, a visually pleasing public open space designed for relaxing will be underused if it sounds unpleasant. Ideally, sonic design should be integrated with visual design, a need the soundscape approach answers. The current trend of co-creating the urban space together with all stakeholders, including local residents, opens up new opportunities to account for all senses in the urban design process. Unfortunately, architects and urban planners struggle to incorporate the soundscape approach in the urban design process and to use it in the context of co-creation. In this work, a hackathon is proposed to generate creative concepts, methods and tools to co-create the urban public space. A soundscape hackathon was organized in the spring of 2019. Participants were challenged to apply their own immersive approaches or virtual and/or augmented reality solutions on selected urban soundscapes. They presented their results to colleagues in the field and to a professional jury. This paper describes the process and results of the event and shows that a hackathon is a viable approach to accelerate the co-creation of the urban public space.

**Keywords:** Hackathon; Soundscape; Immersive; VR; Audio-visual; Co-creation

## 1. Introduction

The creation of new urban environments used to be the responsibility of architects, urban designers and local authorities. Only recently, the role of the citizen has changed to becoming a partner in a co-creation process [1]. This new form of direct participation allows to more directly include the elements that the local population considers as important for liveability of the area [2]. It has been recognized since many years that environmental noise is a main contributing factor to the liveability of an environment [3].

Recently, the awareness has grown that the soundscape of a space is equally important as its visual aesthetics [4,5]. The term soundscape refers to the acoustic environment as perceived by people, in a certain context [6]. Therefore soundscape design aims to (partly) alter that perception or the acoustic environment itself, treating sound as a resource rather than a waste. Not all environmental sounds are noisy or disturbing and should be avoided. Some sounds should be preserved or even accentuated, because a matching soundscape may support the identity and atmosphere (e.g., lively, calming) of the space [7–10].

Sound architects that are involved in the sonic design of urban spaces have a broad range of techniques at their disposal. Examples are in- and out-door auralization models, which aim to recreate

an aural impression of the acoustic characteristics of a space [11], as well as efficient and realistic acoustic simulation, and immersive audio-visual reproduction systems [12]. Virtual reality (VR) and the combination of it with the aforementioned techniques are possibly the most powerful tools currently available. VR is very affordable and not that difficult to work with. It gives the users the impression of being physically present in the environment and allows them to interact with it in real time [13–15]. These techniques are very appealing to both experts and non-experts, and hold new opportunities to motivate urban designers to consider soundscape design, thereby aiming to have a positive impact on the everyday users of the space.

This paper puts forward the concept of a soundscape hackathon, on the one hand as a way to generate tools and methodologies for co-creating the urban sound environment, on the other hand to explore and demonstrate creative ideas for soundscape design. A Soundscape Hackathon was organized as a satellite event to the Urban Sound Symposium, which took place on 3–5 April 2019. The combination of both events attracted acousticians, sound engineers, artists, city representatives, architects as well as urban planners, representing a broad mix of backgrounds that is very valuable in the context of co-creation. The hackathon event took place at De Krook, the new media center in Ghent that houses the city library, imec and different research groups of Ghent University, among which the Institute for Psychoacoustics and Electronic Music (IPEM).

This paper is organised as follows. In Section 2, the general hackathon approach is discussed. In Section 3, the Soundscape Hackathon is described as it was organized by the authors, including task description, team recruitment, information about the provided soundscape data set, the available equipment and the criteria the evaluation was based on. Section 4 then presents the scenarios developed by the different teams. Subsequently, in Section 5, a critical discussion about the hackathon and its goals is presented. Finally, in Section 6, the suitability of the soundscape hackathon as a method to generate creative approaches for improving the urban sound environment in co-operation with various stakeholders is evaluated.

## 2. The Hackathon Approach

### 2.1. Definition

A hackathon can be described as a problem-focused computer programming event [16] or as a contest to pitch, program and present digital innovation prototypes [17]. However, these explanations might not do justice to the type of event a hackathon nowadays has become. The word hackathon is an aggregation of the words hack and marathon. The term hack (or hacking) is not a reference to some kind of malicious cybercrime but is used to express the process of figuring out how a certain system works and subsequently using that information to adapt the existing system to perform previously unintended and unforeseen functions [18]. Based on the etymology of the word, a hackathon can be better described as a social event where participants work together and/or compete to find creative solutions to a challenging problem.

### 2.2. Hackathon Topics

Hackathons are organized in all kinds of fields, having different aims, formats or topics. The online platform hackathon.com [19] lists thousands of hackathon events all over the world, centered around a broad range of topics. They can focus on improving specific applications, certain genres of applications, or on developing new technologies [20]. They can be restricted to participants from specific demographic groups and can even spread beyond the conventional tech world to address social issues [17,20]. Last but not least, company-internal events are organized to encourage new product innovation. For example, Facebook's Like button started as a hackathon project [17,20]. Due to its success, companies and venture capitalists began to see the hackathon as an approach to quickly develop new software technologies, explore new areas for innovation, spot future employees and recruit good ideas worth funding [17,20].

Apart from commercial goals, hackathons can also have scientific aims. Science hackathons focus on bringing researchers together to work on interdisciplinary projects, get new collaborations off the ground or form funding proposals [21]. Even writing a scientific paper can be the goal of a science hackathon, as was the case for the Paper Hackathon on computational life sciences [22,23]. In September 2016, the DREAM of Malaria Hackathon was organized, with the human malaria parasite as a central research topic [24]. This particular event posed a specific research question and aimed towards gaining skills, experience and basic science understanding in its specific field. The cross-disciplinary Sana project [25] has organized hackathons about global mobile health [26]. For them, the hackathon was a way to bring together local geniuses of all sorts, get them to collaborate and develop sustainable solutions based on local problems and needs.

## 2.3. Hackathon Format

Despite the diversity of hackathons, some general conclusions about the format can be drawn. The aims of a hackathon can be defined by the organizers or can be generated at the event [18,20]. In both cases the hackathon starts with some presentations about the event, including information about the challenges, the tools available and practicalities. The work is performed in smaller groups, commonly three to six participants per group. This size of team allows one to work efficiently towards a solution while still having enough different perspectives from the team members [21]. Then, in a strictly limited amount of time [18,20,26] (between one day and one week [20]), teams prepare a presentation and demonstrate their solutions [20]. If the hackathon has a competitive element, a panel of judges selects the winning team and prizes are given to conclude the event [20].

## 2.4. Benefits and Pitfalls

Hackathons are highly interdisciplinary and bring together specialists from different fields to collaborate intensively on a specific problem [18,20]. Different perspectives can generate innovative approaches, which make hackathons the perfect place to accelerate generating outside-the-box ideas [24] and to nurture innovation [20]. Participants develop creative solutions to the problem, improve on communication, teamwork and presentation skills, and transfer their knowledge to team members, which is very valuable from an educational point of view [24]. Last but not least, hackathons are not solely about the results, but also about the relationships between participants with different backgrounds, functions and skill sets, about the opportunity to meet new people and create networks for the long term [20,26].

To obtain creative solutions, participants and especially organizers should take into account some possible pitfalls. Firstly, the competitive aspect can limit creativity, as groups may worry more about other groups rather than focusing on their own work [21]. Secondly, challenging the teams to build a concrete product within a range of restrictions requires a more structured approach, as compared to developing concepts, which in turn may reduce creativity [20,27]. Thirdly, strict time limits require participants to work under time pressure [18,20], which may be counterproductive. To conclude, organizers should be aware that hackathons are not a fast and cheap way to develop apps, software or business plans, but rather a place to get creative ideas flowing and build concepts, which can later be converted into concrete products or designs.

## 3. The Soundscape Hackathon Event

### 3.1. Overview

Its interdisciplinary character makes the hackathon an appropriate tool to benefit from the knowledge and expertise of different parties. To our best knowledge a soundscape hackathon has never been organized, despite the fact that the fields of urban acoustic design and soundscapes encompass a lot of different stakeholders, and that hackathons already have been held to foster the creativity in different domains (see Section 2.2).

The general goal of the Soundscape Hackathon was to redesign and improve the soundscape of urban open spaces. Teams were provided with eight high-quality immersive audiovisual recordings comprising of 360° video and spatial audio. The recordings were collected within the framework of the Urban Soundscapes of the World project (see Section 3.3). Participants were given about 48 h (from April 3rd around noon until April 5th around noon) to complete their work, after which participants of the international Urban Sound Symposium were invited to attend the presentation of the results. Therefore it was important for the teams to think of ways to present their improved scenarios to a public of both experts and lay people. The participants were challenged to *"Do something with the soundscape"*, more specifically to

1. select up to three of the environments given;
2. create a sound environment that enhances the usability of a place and increases its engaging character through a better soundscape [6];
3. assure that their ideas can be implemented and fit in real contexts;
4. create their own tools or to use existing tools to generate the modified audiovisual scenes;
5. use VR, audio rendering and/or auralization to demonstrate their idea to lay people.

## 3.2. Teams

The hackathon was announced on the website of the Urban Sound Symposium and invitations were spread through the academic and professional networks of the organizers. Both individual candidates and teams of up to four people could apply to participate by submitting a short CV and a motivation letter before the end of January 2019. The only information provided to possible candidates was the aim of designing a more suitable soundscape for outdoor public places in a range of cities worldwide and the possibility to win an award in cash. Two individual and six team applications were received, coming from Italy, France, The Netherlands, Austria, Spain and Denmark. Two teams canceled their application and the individual applications were withdrawn because a team of only two members would be too small to efficiently collaborate (see Section 2.3). Therefore, the remaining four teams of acousticians, programmers and artists were selected to participate in the hackathon. Table 1 gives an overview of the four participating teams and their affiliations.

The Italian team 'Immensive' is specialized in computer programming and VR model design. The members of the team founded the start-up company 'Immensive', which develops immersive VR solutions for different fields of applications [28]. Team 'Noize Makers' is a French team with researchers from IFSTTAR (French institute of science and technology for transport, development and networks) and a freelance sound designer/sound engineer. As a team they have experience with audio manipulation, auralization, scientific programming in Matlab and musical software such as Ableton Live. At the time of the hackathon, the members of Team 'Trio Akustiko' were master students at TU Graz. As a trio they performed different scientific and artistic projects, where they gained experience with soundscape recordings, Matlab software development, 3D ambisonics (a surround sound format) and public interactive audio-installations. The members of team 'URCHI' come from different fields and educational backgrounds. Two of the members founded a cultural platform to explore the relationship between the visual and auditory practice. Together as a team, they have experience with software development, music therapy, artistic projects, music composition and performance.

**Table 1.** The selected teams, their country, number of members and affiliation.

| Team | Country | # Members | Affiliation |
| --- | --- | --- | --- |
| Immensive | Italy | 3 | Immensive |
| Noize Makers | France | 4 | IFFSTAR; freelance |
| Trio Akustiko | Austria | 3 | TU Graz |
| URCHI | Spain | 4 | Universitat Pompeu Fabra; Universitat de Barcelona |

### 3.3. Urban Soundscape Data Set

The dataset provided to the teams consisted of a selection of audiovisual recordings, collected within the framework of the Urban Soundscapes of the World (USW) project [12]. The aim of the USW project is to set the scope for a standard on immersive recording and reproduction of urban acoustic environments with soundscape in mind. In this process, a series of documented immersive audiovisual recordings at a range of locations in cities worldwide are collected. The reference database is designed to contain good as well as bad examples of urban acoustic environments, and to support the further introduction of urban soundscape design in education and practice. For a recording from this database to be selected for the hackathon, a reasonably good visual scene has to be available, with a soundscape that could have been better and that leaves room for optimization. Attention was paid to the purpose of the space, the number of people and the sounds present in the scene, and possible salient events that occurred during the recording. In total, eight 3-min recordings were selected, each one performed within a different city. For each of the eight recordings, separate but time-synchronized audio and video files were provided to the hackathon participants. The audio files were recorded with a Core Sound Tetramic first-order ambisonics microphone (including windscreen) and a Tascam DR680 MKII recorder, and were provided as 4-channel 24-bit PCM ambiX files (ACN channel ordering, SN3D normalization) sampled at 48 kHz. Monoscopic 360° video files were recorded with a GoPro Omni, and were provided as equirectangular H264 encoded movies with 4096 × 2048 resolution and a frame rate of 29.97 fps. Table 2 gives an overview of the recording locations, including city, country and coordinates. Figure 1 shows 360° images of the visual scene at the selected locations.

**Table 2.** Overview of the recording locations.

| ID | City (Country) | Location (Coordinates) | YouTube Preview |
|---|---|---|---|
| R0008 | Montreal (CA) | McGill University Campus (45.504202, −73.576833) | https://bit.ly/2Nrj9gu |
| R0018 | Boston (US) | Rose Fitzgerald Kennedy greenway (42.354721, −71.052073) | https://bit.ly/2XyRUo0 |
| R0032 | Tianjin (CN) | Jinwan Plaza (39.131835, 117.202969) | https://bit.ly/2YeMdIZ |
| R0043 | Hong Kong (HK) | Signal Hill Garden (22.296008, 114.174859) | https://bit.ly/2YgrDYx |
| R0063 | Berlin (DE) | Potsdamer Platz Campus (52.509192, 13.376332) | https://bit.ly/2X9NzYV |
| R0064 | New York (US) | City Hall (40.712014, −74.007495) | https://bit.ly/2XEqjS8 |
| R0092 | Chicago (US) | River Walk - Arcade (41.887138, −87.631663) | https://bit.ly/2xcrVUy |
| AT01 | Antwerp (BE) | De Brouwerstraat (51.197695, 4.421701) | https://bit.ly/2Lt24jD |

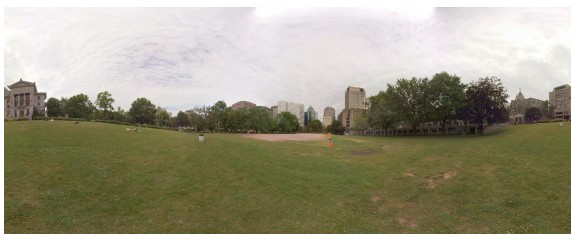

(a)

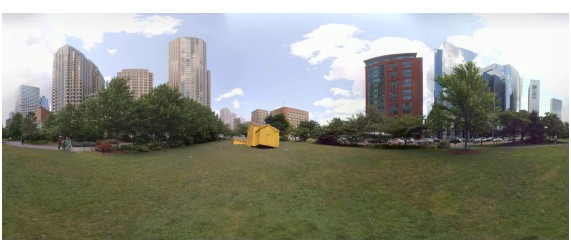

(b)

**Figure 1.** *Cont.*

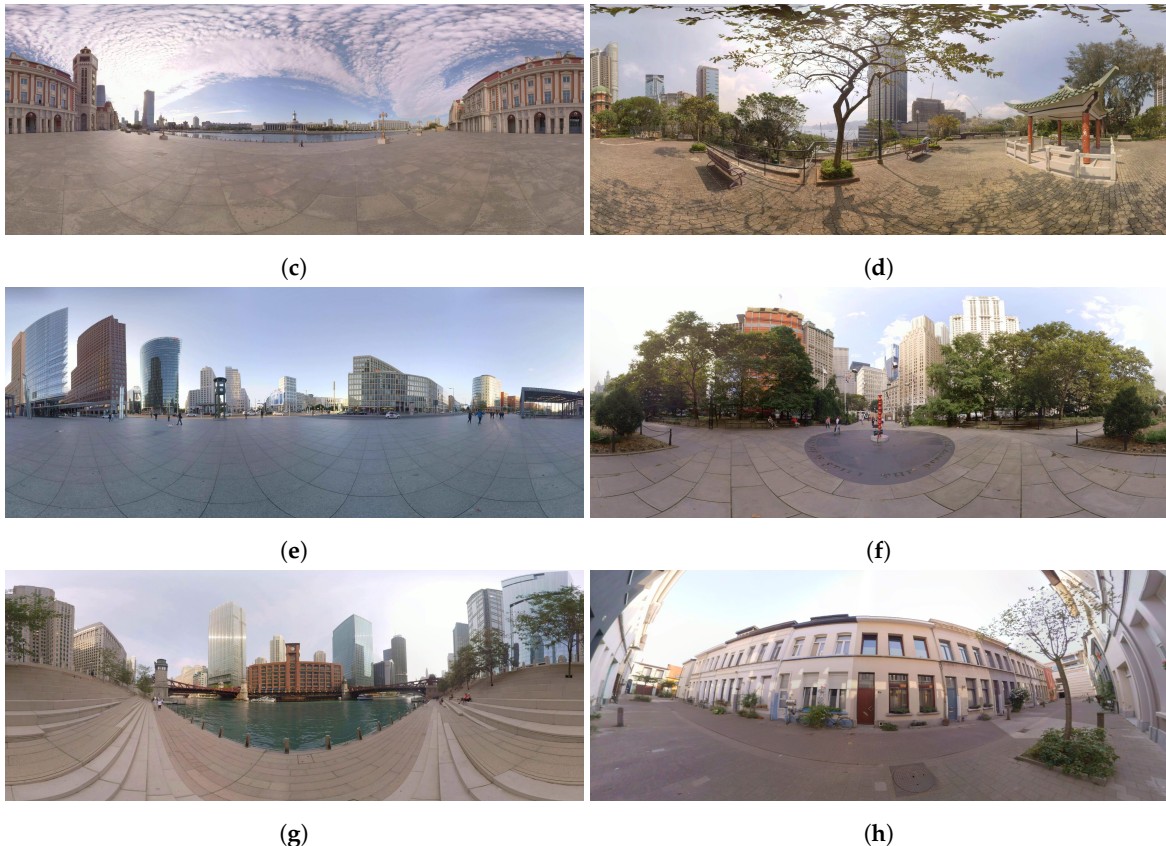

**Figure 1.** The 360°images of the visual scenes. (**a**) R0008: Montreal (**b**) R0018: Boston (**c**) R0032: Tianjin (**d**) R0043: Hong eKong (**e**) R0063: Berlin (**f**) R0064: New York (**g**) R0092: Chicago (**h**) AT01: Antwerp.

The following list describes the purpose and properties of each of the selected sites:

- **R0008** The tranquil lawn on the McGill University Campus serves as a place to relax. Trees and bushes cover the traffic and people that pass by in the background. A constant and monotonous low-frequency noise due to the traffic can be heard, sporadically supplemented by a honking car or whistling birds.

- **R0018** The Rose Fitzgerald Kennedy Greenway in Boston is a 1.6 km long linear park that encourages the sense of a shared community with gardens, promenades, fountains and art installations [29]. The busy area around the location of the recording is reflected in the audio recording by a constant low frequency traffic noise, honking, squeaky brakes, accelerating cars and talking people.

- **R0032** Jinwan Plaza is a spacious square along the borders of the Hai river. The absence of traffic and the calming effect of the water make the spot well suited for a moment to escape from the busy city life. The soundscape mainly consists of low-level noise and some more salient but distant traffic events every once in a while.

- **R0043** Signal Hill Garden is a public park with a lot of natural green, combined with paved pathways, a Chinese pavilion and a panoramic view over Victoria Harbour. Although the garden itself is a calm environment with bird sounds and some periodic noise from garden maintenance, the proximity of industrial cranes and a road creates a rather noisy soundscape.

- **R0063** With around 70.000 visitors a day, Potsdamer Platz is a very lively place and a thriving focal point at the heart of Berlin [30]. Inevitable this is reflected in the soundscape with a lot of talking people and a high amount of traffic with typical sounds such as honking, accelerating, motor sounds etc.

- **R0064** City Hall Park is a small park alongside Broadway with a fountain, art installations and lots of benches, mainly used by tourists to have some rest in between visiting the 9/11 Memorial

and crossing Brooklyn Bridge. Talking people can be heard but the main noise consists of typical traffic noises combined with some ongoing construction works.

- **R0092** The Chicago Riverwalk is a public path along the Chicago River. The shade created by some trees and the presence of water attracts people to have a rest or to make a relaxing walk. Traffic noise and talking people can be perceived, but the recording is dominated by the sound of a tourist boat, including a guide providing touristic information.
- **AT01** The city of Antwerp recently started the project 'Tuinstraten' ('Garden streets') in co-operation with the local community. In such a street the goal is to maximally replace existing pavements with trees, lawns, plant boxes, and other greenery as a measure for climate change and to improve the quality of life in the street [31]. The De Brouwerstraat is a small car-free street where the only thing that can be heard are some background noises and bird sounds.

### 3.4. Equipment

The hackathon event took place in the audio laboratories of IPEM [32]. Participants could use two labs during the event: the 'Maker Space' and the 'Art-Science-Interaction Lab' (ASIL). The next paragraphs describe some of the specific tools and features these labs provide.

### 3.4.1. Art-Science-Interaction Lab (ASIL)

The 'Art-Science-Interaction Lab' measures $10\,m \times 9.5\,m \times 6\,m$ and features a 62 loudspeaker system. The room is acoustically treated to reduce reverberation. In the lower corners, four Martin Audio CX18 sub-woofers are placed; along the walls and ceiling 58 Martin Audio CDD6 loudspeakers are placed, which have a coaxial design and a wide dispersion angle ($110°$). The first loudspeaker ring is located at $2\,m$ height and consists of 34 speakers (cone-to-cone distance $92\,cm$). The second ring is located at $4\,m$ height and has 14 speakers (cone-to-cone distance $184\,cm$). The ceiling array features 10 evenly distributed speakers (cone-to-cone distance $240\,cm$). Figure 2 visualizes the structure. Participants of the hackathon were provided with a list of speaker coordinates in Euclidean (XYZ) and spherical (AED) space [33]. All speakers are powered by Powersoft Ottocanali 4K4+DSP amplifiers, located in an adjacent room. Finally, audio connection to the amplifiers is performed using Audinate's DANTE audio over IP (AoIP) protocol [34]. Participants could connect to the system using one CAT6 Ethernet cable in conjunction with the Dante Virtual Soundcard for 64 discrete audio channel output; or a USB3 soundcard (64-channel RME Digiface Dante interface). Mapping the computer's output channels to the speakers was done in the Dante Controller matrix interface.

### 3.4.2. Maker Space

The 'Maker Space' is an adjacent lab ($20\,m \times 4\,m$) with a smaller 8-channel loudspeaker system placed in a circular array with a radius of $1.8\,m$. Loudspeakers are placed evenly (every $45°$) at $2.5\,m$ height and aimed to the center of the array. The setup is ideal for small-scale individual experiments and tests. The speakers are high-resolution active 2-way studio monitors (Behringer Truth B1030A) connected to a DANTE audio DAC (Focusrite Rednet A16R). Participants could connect to the speakers using the DANTE protocol (over UTP or using USB interface).

### 3.4.3. Audio Rendering Techniques

Two 3D audio rendering techniques were available and recommend to use during the Soundscape Hackathon: wave field synthesis (WFS) and ambisonics. These are physics-based audio reconstruction techniques that aim to create a particular acoustical pressure field at the location of the listener [35] using loudspeaker arrays. Both techniques provide 3D localized sound to reconstruct virtual environments, based on a definition of the room, the audio signal and the desired playback location.

Ambisonics, introduced by Gerzon et al. [36] is based on the decomposition of a sound field into spherical harmonics. Using higher order ambisonics, an enlarged 'sweet spot' can be achieved

for multiple listeners; however, the effect is limited to the center of the room [37]. Wave field field synthesis is achieved through superposition of elementary spherical waves, inspired by the Huygens Principle [38]. The advantage of this technique is the reproduction of physically correct sound fields up to a certain spatial aliasing frequency in an extended sweet spot area, making it suitable for multiple listeners and allowing them to move away from the center [39]. A dedicated WFS sound renderer (Barco IOSONO Core [40]) was available to accurately render different types of 3D sound objects [41,42]: inside or outside the room (point sources) and sources at infinite distance (plane waves). The IOSONO is a user friendly audio processor which translates any incoming audio object (consisting of the audio signal, sound type and playback location) to the discrete speaker outputs. DANTE presets were provided for this configuration. Location data of all audio signals was provided using Open Sound Control (OSC) signals to the IOSONO core. The IOSONO was only available in the ASIL Lab.

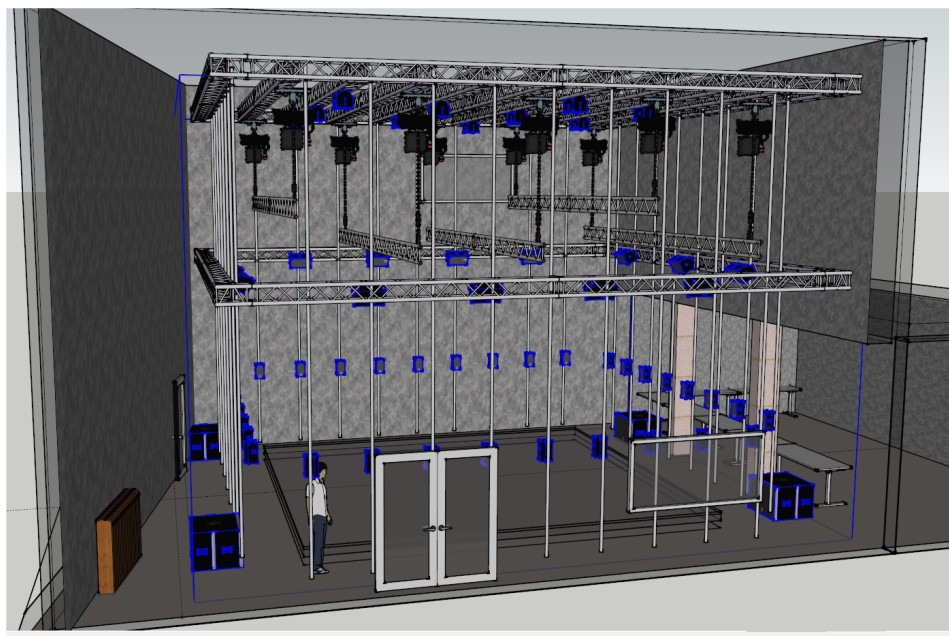

**Figure 2.** Schematic of the Art-Science-Interaction Lab (ASIL). Speakers are accentuated in blue.

### 3.4.4. VR Systems

At the start of the hackathon, each team was provided with an Oculus Go, an easy to use, standalone VR device that can be used with any laptop or smartphone. Together with the device, participants received easy instructions to playback the provided soundscapes on the VR system. To create custom VR experiences, more advanced VR devices are essential and participants could make reservations to use an Oculus Rift, which is more powerful in terms of image quality and immersive experience. The Oculus Rift comes with external sensors, controllers and a pointing device and was provided together with a computer with sufficient requirements (dedicated powerful graphics card). The setup was placed in the Makerspace but was easy to relocate. Reservations could also be made for the HTC Vive Pro, which is even more performing in terms of display resolution, audio quality and tracking accuracy [43]. The accurate tracking sensors of the HTC Vive Pro, capable to track a 10 m × 10 m area, make it very well suited for room-scale immersive experiences [43]. Due to the big tracking area, the HTC Vive Pro was kept in the bigger ASIL lab. To guarantee smooth VR experiences, a powerful computer (a Zotac mini pc with GTX graphics card) was dedicated to the device.

### 3.4.5. Software

Participants were free to use whatever software they preferred. Several suggestions for playback, interactive scenes, audio- and video manipulation were made such as Reaper, Ableton Live 10 [44], some VST plugins, SuperCollider, Blender, Unity etc. Commercial software already available on the

labs computers could be freely used. Next to this, custom code in Matlab, Python, Cycling '74's Max 8 [45], etc. could be created as well.

*3.5. Evaluation Criteria*

The evaluation of the results by the jury was based on three different criteria, which were announced to the participants at the start of the hackathon. All three criteria were equally important and counted for one third of the final score.

1.  Creativity. What concept do the teams use to bring soundscape design to a broad public? Do they use new ideas and concepts, or existing approaches? How do teams cope with the different soundscapes provided? Which one(s) do they select and why?
2.  Theoretical soundness. Do the implemented adjustments sound correct? Is the modification physically possible and realistic? Do the suggested adjustments adhere to soundscape theory?
3.  Use of technology. How do the participants make use of the available technology in their designs? How do they use it to present their ideas? Do they combine different technologies? Is the selected technology suitable to present their idea?

Jury members were selected from the professional and academic soundscape community. To ensure fair competition, attention was paid to their affiliation so that no immediate relations with one of the participating teams existed. Based on their experience and academic background all three evaluation criteria were covered. The jury consisted of professionals affiliated with Leiden University, the Netherlands; UCL, United Kingdom; TU Eindhoven, the Netherlands; McGill University, Canada; alioop.com, Canada; Ghent University, Belgium and ASAsense, Belgium.

*3.6. Timeline*

On the first day, the participants arrived in the morning to attend presentations about the goal of the hackathon. A detailed task description was provided, as well as information about the audiovisual recordings, the available infrastructure and the criteria the jury would base their evaluation on. From then on, the participants were free to schedule their time as they pleased. The teams had the possibility to make reservations for 2-h time slots to use the available equipment (see Section 3.4). One team used their own VR equipment while other teams agreed to share the available lab space, for example, one team used the audio hardware in the lab itself while another team used the control room to work with the VR software. Therefore all teams could reserve multiple time slots per day to make the most out of the equipment that was offered. The third day, teams had a final half day to wrap up their work, setup hardware and software for a demonstration and prepare presentations. After presenting to the jury and the visitors from the symposium, followed by a long jury deliberation, the hackathon came to an end with an award ceremony and an informal reception.

## 4. Results

The four teams had a different approach to developing a modified soundscape scenario. Some focused on the visual scene, others improved the spatial audio recordings, developed interactive scenarios, or presented a custom framework. The following sections summarize the approach and results of each of the teams.

*4.1. Immensive*

Team Immensive selected the McGill University campus in Montreal to work with because of the characteristics of its sound environment. The ambient sound is rather quiet and there is little disturbance from other noise sources. This creates opportunities to enhance the usability of the place by carrying out interventions on different areas inside the park. Two different areas were selected for two different use-cases: a fitness island towards the center of the lawn and an arc-shaped relaxing zone that surrounds the fitness island. The fitness island consisted of three gym exercise tools: a rowing

machine, an elliptical machine and an exercise bike (Figure 3a). The fitness machines aimed to make the place more attractive and to enhance the sound environment with the sounds of the related real activities: respectively paddling in the water, cross-country skiing and the sound of wheels, pedaling and wind. A second aim was to make the gym activities more enjoyable by having sounds from real-life activities, allowing to omit the use of earphones. The relaxing zone is created using sonic benches emitting natural water sounds (Figure 3b). Users can interact with the benches through their smartphones to manage the type of the sound and the intensity level. These 'intelligent' benches allow to adapt the soundscape to the people's desire.

Team Immensive presented their idea in VR, using an Oculus Rift, where the user can 'row' on the rowing machine or experience the sonic benches. All software was created by the team itself. The provided ambient ambisonics recording was mapped to a virtual 7.0 surround loudspeaker setup, whereas the added sounds were modeled as point sources with hemispheric sound propagation. Both ambient and modeled sounds were combined in the virtual scene using the Unreal game engine.

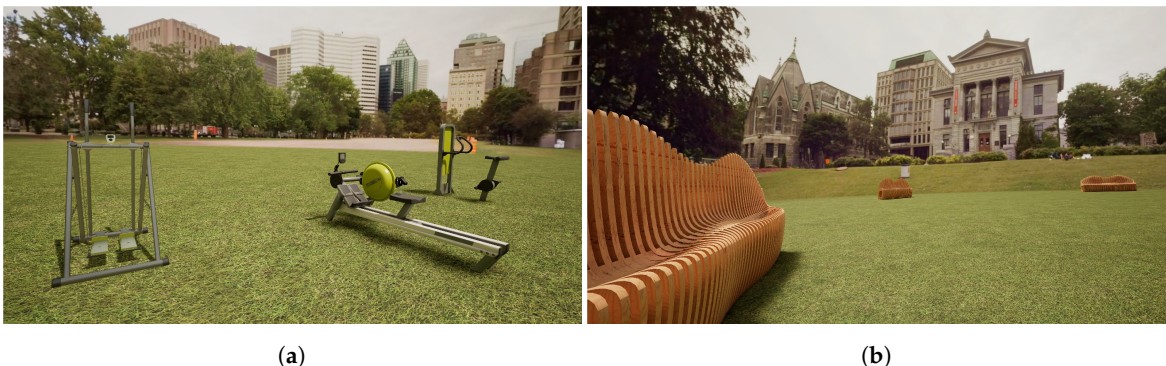

| (**a**) | (**b**) |

**Figure 3.** Screenshots of the two use-cases of team Immensive to enhance the place's usability. (**a**) Fitness island with three gym exercises including matching sounds. (**b**) Relaxing zone with sonic benches.

### 4.2. Noize Makers

Team Noize Makers start their presentation with a video on a television screen. The video zooms in on a map of New York and guides the spectators to the location of the selected soundscape: New York City Hall Park. A voice-over explains why this is the most complete situation with the greatest potential to modify the soundscape: because several types of sounds are present in the scene (cars, construction works, horns, water and voices). Together, these sounds create a noisy and polyrhythmic soundscape with little space for relaxation in or contemplation of the city. Ambisonic playback of the ambient sound through the surrounding speakers in the ASIL lab makes very clear that although being in a park, the heavy rhythm of New York predominates. Two questions are posed: "How can we stop for a moment the city clockwork from running so fast?", and "How can we bring back for an instant the harmony in the city?". Before the team tries to give an answer to these questions by means of their modified scene, the voice-over reads the poem 'Heard' from Ellen Reiss, describing the feeling of a local when hearing New York City.

In the second part of the presentation, the focus is shifted towards the virtual scene through the VR goggles. Sounds still come from the surrounding speakers in the ASIL lab. Wave Field Synthesis (WFS) is used to add very localized sound objects to the ambient noise of the scene. Usually all 62 speakers in the ASIL lab are used for WFS, because they are configured to reach optimal performance. This implies that no speakers are available to play the ambisonics recording of the ambient sound. The team implemented a clever hack to circumvent this problem and used only part of the speakers for WFS, while the other part was used to playback the ambisonics recording. Careful speaker selection limited the degradation in the WFS effect, possibly leading to a better combined WFS and ambisonics effect. A fountain, flags, different birds, voices, sculpture noise and a progressive climbing drone were added

as sound objects. In the visual scene, these added sounds were emphasized by means of 3D legend objects, pointing towards the location of the sound (Figure 4).

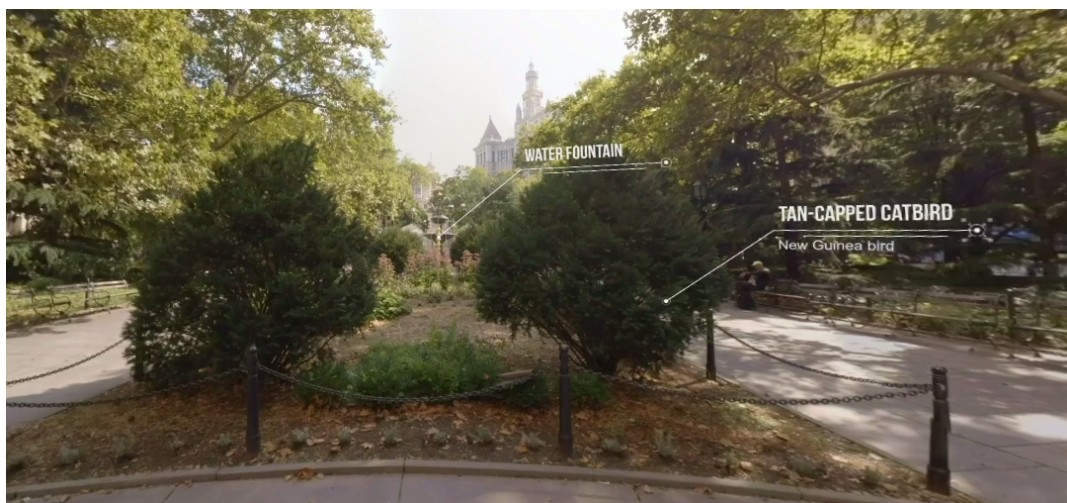

**Figure 4.** Screenshot of the demonstration by team Noize Makers, using 3D legend objects that point towards the location of the localized sound objects.

### 4.3. Trio Akustiko

Team Trio Akustiko selected the Potsdamer Platz soundscape in Berlin. The open space on the big square allowed them to add an extra virtual layer to the visual scene. Through a VR headset, the user can experience the added layer in the center of the scene, with the original recording shown in the background (Figure 5). The team added five buttons in the virtual scene: 'Fountain', 'Trees', 'Children', 'Cafe' and 'Hills' (Figure 5c). Clicking these buttons adds a fountain, a series of trees, playing children, a food-truck serving as a pop-up bar and some low hills with benches to sit on. Users can adjust the visual scene as they wish, with the elements or combinations of elements they prefer. Not only the visual scene is adjusted, as adding those elements also adds corresponding sounds to the specific location of the virtual objects. The original soundscape of Potsdamer Platz can be heard in the background. Ambisonics recordings were used to play through the speaker system in the ASIL lab, while stereo sounds were played back through open-ear headphones (Sennheiser HD650). The approach of Trio Akustiko allows people to really experience how the improved urban design could be, both in a visual and in an acoustic way. Multiple options can be explored and people can form their opinion on the different designs by immersing themselves in the virtual scenes.

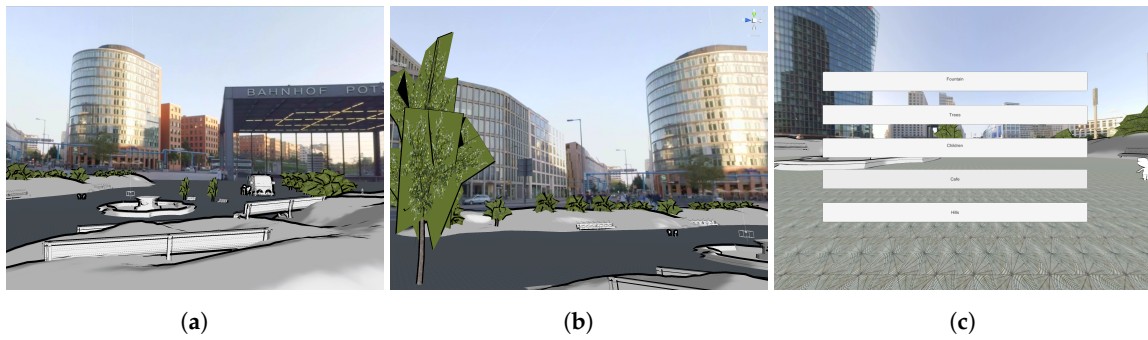

|        (**a**)        |        (**b**)        |        (**c**)        |

**Figure 5.** Screenshots of the virtual layer added to Potsdamer Platz by team Trio Akustiko. (**a**,**b**) Different perspectives on the virtual scene. (**c**) Buttons allow modification of the scene.

### 4.4. URCHI

Team URCHI developed two different scenarios for two different soundscapes. Their ideas were explained with a visual presentation followed by an immersive audio experience. First, the team

selected the Signal Hill Garden recording for creating a restorative soundscape. A positive acoustic design intervention (adding sounds) was created in order to generate comfort and quietness. The team was inspired by the Chinese pavilion in the scene and they modified typical Chinese lanterns with wind chimes. Wind harps were added as well because together they have a relaxing and meditative effect that creates peace of mind and a sense of harmony. Figure 6 shows the proposed adaptations and a sketch of the adapted scene. Important to note is that the adaptations were only performed in the auditory scene; nothing was changed in the video recording.

Customized SuperCollider software was developed to synthesize the sound of the wind chimes in three octaves tuned to a microtonal Makam Rast scale. Traditionally, it is believed that such a scale elicits comfort, releases stress and brings tranquility. A wind harp sample was ambisonics encoded and spatially placed in elevated corners of the scene. Movement of wind was simulated by consecutive fading in and out of a wind sound sample. Entering the garden is simulated by applying a spatial filter towards the back of the scene (the water in this case) at the beginning of the recording. To make everything sound natural and more immersive, the sound levels of the different soundscape components were manually adjusted.

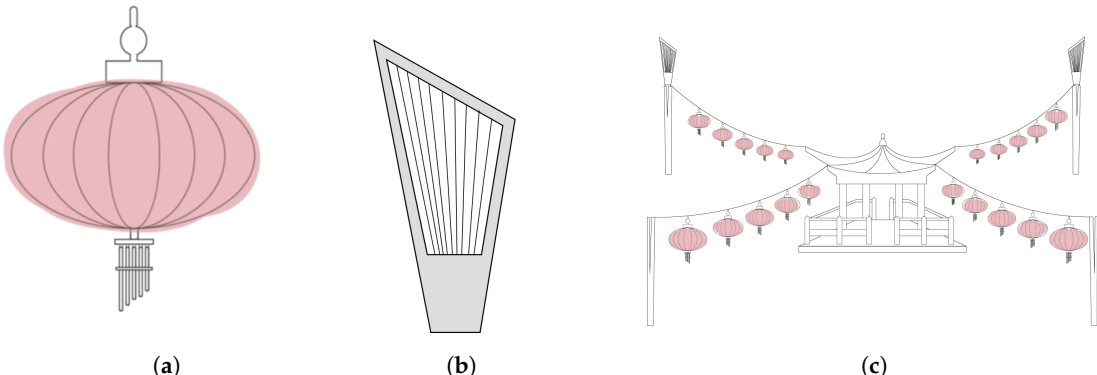

(**a**)　　　　　　　　　　　　　(**b**)　　　　　　　　　　　　　(**c**)

**Figure 6.** Adaptations to Signal Hill Garden by team URCHI. (**a**) Typical Chinese lantern modified with wind chimes. (**b**) Sketch of a wind harp. (**c**) Sketch of the adapted scene.

Subsequently, the team selected the Boston soundscape that is characterized by traffic noise, car horns, tram bells and voices. To improve this soundscape, the team applied both a positive (adding sounds) and negative (removing/hiding sounds) acoustic design. The idea was to place a dome overgrown with ivy, a creeper, on the lawn of the Boston park to screen the surrounding noise [46,47]. A small fountain was then placed inside the dome, as the sound of water is known to be an effective natural sound to mask road traffic noise [48,49]. Water sounds also increase the overall impression of pleasantness, eventfulness and perceived quietness of the soundscape [49]. Finally, the team placed a gravel path leading to the entrance of the dome, with associated sounds having the effect of attracting attention. Figure 7 shows a sketch of the intervention.

To create the effect of the dome in the audio recording, a fitting lowpass spectral filter was designed. A person walking towards the dome is simulated by playing a sample of calm gravel steps and placing it in the 3D scene. Similarly, the fountain is placed in the scene, with the designed filter applied to the fountain sound to create the impression that it is inside the dome. Upon entering the dome, the filter is removed from the fountain sound while simultaneously being applied to the surrounding soundscape recording. By changing the spatial position of the fountain, moving inside the dome is simulated. Again, sound levels of the different soundscape components were manually adjusted to create a natural sounding and immersive experience.

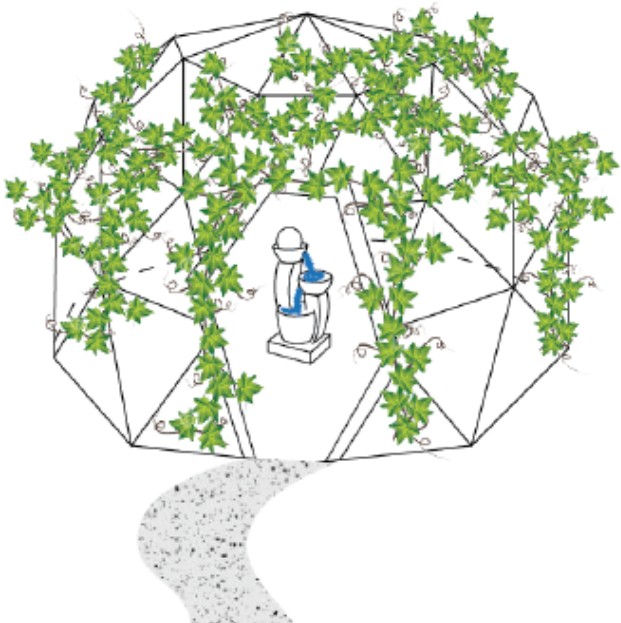

**Figure 7.** Sketch of the Boston intervention by team URCHI.

## 5. Discussion

### 5.1. Hackathon Outcomes

The main goal of including creativity in the evaluation criteria was to promote outside-the-box thinking in generating creative ideas for more fitting soundscapes. A common thread in all presented interventions is the use of green elements, or more generally, elements from nature. Green spaces in particular are attractive for several reasons: they encourage social interaction, stimulate physical activity and give opportunities for mental restoration [50]. Moreover, the beneficial effect of green elements on the urban soundscape is well documented in literature, see for example [7,51]. Although all presented interventions focus in some way on nature-related aspects, the four teams took a different conceptual approach. Team Noize Makers emphasized and improved existing elements, for instance bird sounds or a fountain in the background. Teams URCHI and Trio Akustiko added additional green elements, such as a dome overgrown with ivy (URCHI), or even an entire scene with hills and trees (Trio Akustiko). Three teams gave materialized support for the services the green provides: wind chimes or a fountain for relaxing sounds (URCHI, Trio Akustiko), a cafe for social interaction (Trio Akustiko), resting benches for mental restoration (Trio Akustiko, Immensive) or gym exercise equipment for stimulating physical activity (Immensive). One could consider the gym stand to be superfluous, since the park itself gives plenty of opportunities for performing similar excercises. However, these installations add the potential of social interaction to the physical stimulation, as it may function as an attraction point in the park. The presence of green elements is strongly related to the tranquility of a space. In general, three groups of people can be discerned based on how they attach meaning to the concept of tranquility. They either associate it with silence, natural sounds or social relationships [52]. Approaches that emphasize existing green elements (Noize Makers), or that involve adding new green elements (URCHI), address the need for natural sounds and to a lesser extent also the need for silence. Approaches that support the services for social interaction that the greenery provides (Immensive, Trio Akustiko), address the importance of social relationships. Although it is not required that all viewpoints on tranquility are covered in each urban place, it would be advantageous that all teams in a hackathon consider them. To this end, it is necessary to have a well balanced team with participants covering these different viewpoints.

Theoretical soundness was included in the evaluation criteria in order to counterbalance creativity, as it should be physically feasible to implement those creative ideas in the urban scene. Additionally

they should make sense, in order to effectively increase the quality of the soundscape. All teams spent a considerable amount of effort to make their designs sound plausible. However, the extent to which the solutions could be implemented in real life varies considerably. Interventions such as those in New York City Hall Park (Noize Makers) are meant to be implemented on top of existing scenery, without physically altering the real environment, and could be relatively easily implemented. Other interventions, such as those at Potsdamer Platz (Trio Akustiko) or at the Rose Fitzgerald Kennedy greenway in Boston (URCHI) would require considerable landscaping efforts to realize. Moreover, they would require considerable changes to the original design in order to make sense acoustically because the acoustical screening effects are overestimated in the initial auralization. This illustrates some of the pitfalls of co-creation: firstly, participants may violate the laws of physics due to a lack of knowledge; secondly, some factors might be neglected, for instance the cost of implementing a new design; thirdly, some indirect effects, e.g., the flow of persons in emergency situations, may not be considered adequately due to lack of expert knowledge in particular fields.

The use of technology as an evaluation criterium assesses the way the teams made use of the available tools to present their ideas in a convincing way. As discussed in Section 3.4, the hackathon participants had a wide range of visualization and audio rendering techniques at their disposal for presenting their solutions. The results of the hackathon show that (a combination of) the current technologies allow to audio-visually present the redesign of a space in a limited time frame of two days and with relatively good quality.

On the one hand, most interventions revolved around adding sounds, which is reflected in the dominant choice of rather quiet (e.g., Montreal McGill University Campus) or relatively confined spaces (e.g., New York City Hall Park or Hong Kong Signal Hill Garden). Added sounds are sometimes visually accentuated during the presentation, such as in the intervention in New York City Hall Park (Noize Makers), where special legend objects guide attention to selected interventions. For the interventions in Potsdamer Platz (Trio Akustiko) and the McGill University Campus (Immensive), accentuation during presentation is less critical, as the added objects are rendered differently on top of the original visual scene and already focus attention. However, these objects may lose appreciation of their added functionality since they visually dominate the scenery. As a result, having a balance between attracting attention and becoming part of the scenery is an important rendering aspect for creating the VR outcome. On the other hand, two interventions involved suppressing existing sounds: the added hills at the Potsdamer Platz (Trio Akustiko), and the dome at the Boston greenway (URCHI). With most audio rendering software or VR engines, new sounds can be added relatively easily to an existing soundscape. Suppressing existing sounds originating from a particular direction poses a greater technical challenge. Given the limited time available, both teams provided a first impression of the effect of their intervention by reducing and spectrally shaping the ambient soundscape as a whole, approximating the physics of sound propagation involved. Nevertheless, in order to allow participants to investigate the full potential of suppressing existing sounds, future soundscape hackathons should consider providing easy-to-use software tools for sound propagation, such that participants can focus on the creative aspect rather than on physics.

Another common aspect in all interventions, particularly made possible through the use of VR technology, is the interaction between the user and its environment. Interaction, where the user of a space can to some extent take control of its soundscape, creates a larger sense of presence and immersion, and therefore may make the presentation more convincing. Basic forms of interaction with which the user can shape its soundscape were included in several interventions, such as the sound of footsteps when walking in the Boston greenway (URCHI) or the sound of the gym tools (e.g., the moving water of the rowing machine) at the McGill University Campus (Immensive). In more advanced forms of interaction, users are able to manipulate objects inside the VR environment in order to create sound or alter the soundscape. This requires haptic feedback devices, which were not available in this hackathon. Future soundscape hackathons would therefore benefit from having

such tools at the disposal, allowing participants to design sound environments with a higher level of interactivity.

*5.2. Evaluation of the Event*

As this was the first soundscape hackathon, the authors acknowledge that there is still room to improve the organization of the event. To receive feedback and to gain more insight into the organization of the event, a small questionnaire for evaluation was distributed among participants, jury members and those attendees of the Urban Sound Symposium that registered to attend the final presentations and demonstrations. The questionnaire included a mix of rating questions and open-ended question and dealt with, among other topics, format, general appreciation, and suggestions on improvements. Because the number of respondents was small, the questionnaire was not used as a statistic way to evaluate the event, but rather as an inspiration for improvement. In general, the event was perceived as creative and interesting, and certainly to be repeated. Combined with how the organizers experienced the event, three main points of improvement could be extracted from the questionnaire responses.

A first point of improvement is the selection of participants. As the event was inherently connected to the Urban Sound Symposium and was shared among the academic and professional network of the organizers, participants mainly were acousticians and sound professionals with a technical background. According to the respondents, other parties should be invited as well, including artists, architects, city representatives, public space designers, software developers and residents (in order of importance as stated by the respondents). Given a wider mix of participant backgrounds, more creative ideas could have emerged.

Secondly, due to the organization of the event in parallel to the Urban Sound Symposium, the jury members were not able to visit the teams during the hackathon itself. To decide on a winner for the Soundscape Hackathon, the jury members individually gave points on the three evaluation criteria to obtain an overall score for each team, thus making it an outcome-centered evaluation. Several respondents suggested to have multiple smaller awards for different sub-challenges, instead of one team winning everything. This could reduce the competitive nature of the event and instead stimulate cooperation, possibly leading to better results. One could assess the degree of collaboration within and between the teams as an extra criterion or as a sub-challenge, next to the quality of the work. In the outcome-centered evaluation as it was, Team Noize Makers was declared as the winning team and received a trophy and a monetary award.

Thirdly, the organization and timing of the final presentations can definitively be improved. Given the short duration available for the final presentations and the use of the same equipment by different teams, it was hard to coordinate efficiently and to provide access to a broader audience. However, in context of co-creation, the presence of and the interaction with a broader audience can reflect the participation of local residents. Even an award based on the score of the audience could be included, as suggested by the respondents, instead of only having a jury to provide scores.

The Soundscape Hackathon was organized to explore the potential of the hackathon format in a co-creation context for generating creative concepts, methods and tools to increase the quality of the soundscape of urban outdoor spaces. To achieve this goal, the aforementioned improvements must be taken into account. The hackathon should bring together a wide mix of stakeholders to enhance their project participation. Evaluation should be based not only on the outcome of the presented work, but the interaction of the hackathon teams with each other and with the different stakeholders will have to be integrated in the hackathon process, and should be taken into account in the evaluation. In this case, the composition of the jury will need to reflect the different stakeholders and will need to be extended with moderating professionals. When also a broad audience of local residents can attend final presentations or can even actively participate, new opportunities for co-creation arise.

## 6. Conclusions

This paper investigated the hackathon format as a way to generate ideas and creative concepts for applying the soundscape approach in urban public space design. In a typical hackathon, people from different fields participate, applying their own specific knowledge and expertise to the topic. This mix of backgrounds and points of view may create innovative approaches and accelerates outside-the-box ideas. To test this approach, a Soundscape Hackathon was organized in Ghent, Belgium on 3–5 April 2019. The participants of the hackathon were challenged to design a series of urban soundscape interventions, to apply them using a range of virtual reality visualization and auralization technologies that were available at the hackathon venue, and finally to present their solutions to colleagues in the field and to a professional jury. This paper described the process and results of the event, and discussed the benefits and shortcomings of a soundscape hackathon. Such an event not only creates creative and interactive concepts, methods or tools in context of co-creation, but when a broad mix of participants with different backgrounds is selected, it can be a reflection of all stakeholders involved in the co-creation process. It is even possible to include a broad audience in the event, extending the co-creation idea even further. Given that this was the first soundscape hackathon edition, some practical shortcomings such as the organization of the final presentations were to be expected. By taking into account that participants can possibly oversimplify important aspects such as physical soundness, safety or cost on the one hand, or that their creativity can be limited by strict time constraints or the aspect of competition on the other hand, the hackathon methodology can still be improved. Both participants and organizers perceived the event as successful, and found it interesting to see how young scientists and young professionals came up with creative solutions, immersive approaches and artistic presentations. Although the creativity of the participants was as expected and they exclusively used the provided technologies, the organizers were surprised and impressed by the level of the teams' solutions, especially given the short time frame. All in all, this first soundscape hackathon edition showed that the format and main concepts of a hackathon are well suited to be applied in urban sound design, and that the format may present a viable approach for accelerating the co-creation of the urban public space.

**Author Contributions:** Conceptualization, P.D., D.B. and B.D.C.; methodology, J.D.W., K.F., B.M., P.D., D.B and B.D.C.; formal analysis, J.D.W., K.F., B.M., P.D., D.B. and B.D.C.; investigation, J.D.W., K.F. and B.M.; resources, M.L., D.B. and B.D.C.; writing–original draft preparation, J.D.W., B.M. and B.D.C.; writing–review and editing, J.D.W., K.F., B.M., P.D., D.B. and B.D.C.; visualization, J.D.W.; supervision, P.D., M.L., D.B. and B.D.C.; project administration, D.B. and B.D.C.; funding acquisition, B.D.C. All authors have read and agreed to the published version of the manuscript.

**Funding:** This research was funded by the HEAD Genuit Foundation grant number P-16/11-W "Urban Soundscapes of the World" . The support of this foundation is gratefully acknowledged. The research was part of C3PLACES ("Using ICT for co-creation of inclusive public spaces"), and received funding from the European Union's Horizon 2020 research and innovation program (Urban Europe) under grant agreement no. 693443.

**Acknowledgments:** The authors want to thank all participants and jury members of the first Soundscape Hackathon.

**Conflicts of Interest:** The authors declare no conflict of interest. The funders had no role in the design of the study; in the collection, analyses, or interpretation of data; in the writing of the manuscript, or in the decision to publish the results.

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
