# Peer review of "The Soundscape Hackathon as a Methodology to Accelerate Co-Creation of the Urban Public Space"

_applsci, doi:10.3390/app10061932_

Round 1

Reviewer 1 Report

I found the proposal original and very interesting, and it perfectly fits the scope of the Special Issue. Therefore, I would recommend the paper to be published in the Special Issue "Soundscape in Architecture and Urban Planning" but after minor corrections that should mainly focus on the improvement of the discussion and the conclusion sections.

In general, the manuscript is clearly written, the methodology is thoroughly described and the references used are adequate. The motivation and the purpose of the work are clearly described in section 1. Section 2 is too long in my opinion, including some details and examples which are not necessary, such as all the listed examples from previous works where the reference it is enough (See lines 75 "such as women, teenagers or college students"; or line 109 "(e.g., engineers, artists, scientists, policy makers); among others). Also in this section, the sentence "To obtain creative solutions, maximal creativity should be encourage and support" (lines 112-113) is a bit confusing, I think is used as an introduction to the following ideas, but as it is the sentence adds nothing to the paragraph, so please, rephrase.

In section 3.2, I found a couple of minor issues. The reason why only 4 teams were selected for the hackathon is not very clear. A priori, having 6 teams is better than having 4 involved, so readers may wonder why this decision was made. Some explanation is given in line 161 but is very vague. I would recommend including the reasons why 2 of the teams were discarded, especially in case any of these criteria could be used in the future and having into consideration during the recruitment process. Figure 1 is not needed. First paragraph in section 3.4.3 it is confusing, please rephrase. In line 317, I would delete the sentence "Final presentations were scheduled to start after lunch, so". Section 4 is clear.

I think sections 5 and 6 must be improved so that the paper is not only a report of the event but an example of a method or format that can be use " as a way to generate ideas and creative concepts for applying the soundscape approach in urban public space design" as stated at the beginning of section 6. As they are, the discussion of the results are weak in comparison with how the methodology and the results are described.

The first part of section 5.1, where the discussion is based on the evaluation criteria is more a summary than a discussion, so I think this part need to be improved adding some critical comments to the teams' proposals and a further analysis on, for example, how the background of each team may influence their decisions (exactly as authors did in lines 451-455). Paragraph 1 and 2 should be just one, information in brackets in lines 426 and 427 is no needed, and referring to "some" or "others" teams when there are only 4 and are then listed is not needed either. Sentence in lines 461-462 is a repetition, so I suggest deleting or rephrasing it. The second part of the section, from line 463 to 486 is much better. I suggest extending this part, so that some guidelines can be formulated with scientific soundness, which I think is the real potential of this work.
In section 5.2 lines from 488 to 499 are a bit repetitive, so I don't think they are needed. I would also suggest that authors reconsider the title of this section, since it is more a final evaluation of the event. If the survey is used as a method for evaluation, I would include a bit more information about it, such as the number of respondents, range of ages, backgrounds, if it used open-ended questions or rating questions.... I suggest that authors used all this information to formulate some guidelines or tips to be considered for future events of this nature, from the organization to the evaluation processes.

Conclusion section is too general, I suggest that authors include a summary of the benefits and shortcomings of using this format (Line 538)

Reviewer 2 Report

Overall a good paper laying out a clear methodology and motivation.

The concept and purpose of a hackathon is well established, and it is good to see this process used in an explicitly soundscape based context.

It would be interesting to see some comment on whether the actions of the four teams that took part were similar to what the organisers expected.

Overall the paper is well written and clearly presented, although I think section 5 needs some work in terms of presenting the outcomes in a mcuh clearer way, and making more explicit how the results presented in this paper could be integrated into future urban design etc. The paper as it is does do this to a degree, but I think the presentation could be clearer in this regard.
